# Immunohistochemical Expression (IE) of Oestrogen Receptors in the Intestines of Prepubertal Gilts Exposed to Zearalenone

**DOI:** 10.3390/toxins15020122

**Published:** 2023-02-02

**Authors:** Magdalena Gajęcka, Iwona Otrocka-Domagała, Paweł Brzuzan, Michał Dąbrowski, Sylwia Lisieska-Żołnierczyk, Łukasz Zielonka, Maciej Tadeusz Gajęcki

**Affiliations:** 1Department of Veterinary Prevention and Feed Hygiene, Faculty of Veterinary Medicine, University of Warmia and Mazury in Olsztyn, Oczapowskiego 13, 10-718 Olsztyn, Poland; 2Department of Pathological Anatomy, Faculty of Veterinary Medicine, University of Warmia and Mazury in Olsztyn, Oczapowskiego 13, 10-718 Olsztyn, Poland; 3Department of Environmental Biotechnology, Faculty of Environmental Sciences and Fisheries, University of Warmia and Mazury in Olsztyn, Oczapowskiego 13, 10-718 Olsztyn, Poland; 4Independent Public Health Care Centre of the Ministry of the Interior and Administration, and the Warmia and Mazury Oncology Centre in Olsztyn, al. Wojska Polskiego 37, 10-228 Olsztyn, Poland

**Keywords:** zearalenone, immunohistochemistry, oestrogen receptors, gilts before puberty

## Abstract

This study was conducted to determine if a low monotonic dose of zearalenone (ZEN) affects the immunohistochemical expression (IE) of oestrogen receptor alpha (ERα) and oestrogen receptor beta (ERβ) in the intestines of sexually immature gilts. Group C (control group; n = 18) gilts were given a placebo. Group E (experimental group; n = 18) gilts were dosed orally with 40 μg ZEN /kg body weight (BW), each day before morning feeding. Samples of intestinal tissue were collected post-mortem six times. The samples were stained to analyse the IE of ERα and Erβ in the scanned slides. The strongest response was observed in ERα in the duodenum (90.387—average % of cells with ERα expression) and in ERβ in the descending colon (84.329—average % of cells with ERβ expression); the opposite response was recorded in the caecum (2.484—average % of cells with ERα expression) and the ascending colon (2.448—average % of cells with ERα expression); on the first two dates of exposure, the digestive tract had to adapt to ZEN in feed. The results of this study, supported by a mechanistic interpretation of previous research findings, suggest that ZEN performs numerous functions in the digestive tract.

## 1. Introduction

Oestrogens and oestrogen-like substances found in the natural environment including the mycoestrogen ZEN, affect the developing reproductive and non-reproductive tissues [1,2]. Oestrogens are synthesised by the body, but they are also present in the environment, in the form of xenobiotics and naturally occurring compounds (undesirable substances) [3]. Most of these substances (not necessarily pollutants) are known as endocrine disruptors (EDs) [4], and they are usually found in soil, air, water, food and feed (i.e., the environment) [5,6]. Phytoestrogens (genistein, coumestrol) and the mycoestrogen ZEN (fungal metabolite) are naturally occurring EDs [7,8,9].

Zearalenone and α-zearalenol (α-ZEL) have an oestrogen-like structure. However, they are not steroids and do not originate from sterane structures [10]. EDs such as zearalenone are involved in several processes [11,12] that influence the endocrine system [13] and induce side effects [14]: (i) in prepubertal gilts, EDs compete with endogenous oestrogens for the binding sites of oestrogen receptors (ERs), which can alter mRNA expression levels and protein synthesis and reduce the efficacy of endogenous steroids [10,15,16,17]; (ii) EDs can bind to the inactive receptor (i.e., blocking it), thereby preventing the binding of natural hormones to that receptor (antagonistic effect) [11,17]; (iii) EDs reduce the levels of circulating natural hormones because they bind to blood transporting proteins, [2]; and (iv) EDs can also affect the body’s metabolism by influencing the rates of synthesis, decomposition, and release of natural hormones [10,18,19,20].

When ingested, ZEN can prevent or delay the clinical and subclinical spread of oestrogen-dependent tumours [2,21,22]. Sex hormones and exogenous oestrogen-like chemicals are frequently implicated in the aetiology of tumours in various tissues [8]. Many oestrogen-sensitive tumours are termed oestrogen receptor-positive tumours because ERs are mediators of oestrogens or oestrogen-like substances that cause cancer [14,21]. Zearalenone may be a selective oestrogen receptor modulator, but its binding affinity for ERs is 10,000 times lower than that of 17-oestradiol (E_2_) [2]. Zearalenone has agonistic or antagonistic effects on target tissues, depending on the type of ER [1,2]. The chemopreventive effect of ZEN can be attributed to its antagonistic influence on ERs [18]. There is evidence that ZEN can inhibit circulating oestrogen precursors and slow the development and progression of oestrogen-dependent tumours by binding to ERs, and ERs can probably also inhibit the activity of steroid hormones that convert circulating hormones to E_2_ [18,23]. 

Elements of the oestrogen response have been investigated in studies involving endogenous oestrogens and oestrogen-containing drugs [12,13,18]. When endogenous oestrogens exert genomic effects via ERs, oestrogen response elements bind with ERs or other response elements in the neighbouring genes that respond directly to oestrogens [3]. The resulting bonds influence the transcription of oestrogen-responsive genes. Mycoestrogens trigger similar responses by binding to ERs and initiating molecular cascades that alter gene expression [8]. Zearalenone is involved in molecular mechanisms, but its oestrogenic activity remains insufficiently investigated. Previous research has demonstrated that the presence of ZEN in feed or food affects the mRNA expression of ERs [8,24] and the activity of other genes encoding metabolic processes in enterocytes [25,26]. Subclinical symptoms of ZEN mycotoxicosis can cause changes in hormonal signalling when enterocytes in different intestinal segments are exposed to this mycotoxin [19]. The role of zearalenone in the digestive system should be evaluated to determine possible risks for gilts before puberty [2,27,28,29,30]. Therefore, this experiment aimed to find out whether a low monotonic dose of ZEN affects the immunohistochemical expression (IE) of ERα and ERβ in the gut of prepubertal gilts. The findings may contribute to a mechanistic understanding of changes in ERα and ERβ expression.

## 2. Results

### 2.1. Clinical Observations

Clinical manifestations of ZEN mycotoxicosis were not noted during the experiment. However, histopathological analyses, ultrastructural analyses, and analyses of the metabolic profile of samples taken from same gilts frequently revealed changes in certain tissues or cells. These findings have been posted in various articles [2,19,20,31,32,33,34,35].

### 2.2. Optical Density

The brown background staining of the slides (Figure 1 and Figure 2) was not specific to all intestinal segments, and it may have occurred in staining assays examining the ERα and ERβ expression in DAB-stained gastrointestinal tissues (most samples exhibited light-brown, non-specific staining).

The effect of six-week exposure to ZEN on the expression levels of the selected ERs was determined in selected segments of the gastrointestinal tract (GI) of gilts in the control and experimental groups using a four point scale (negative—0; weak and homogeneous—1; mild or moderate and homogeneous—2; intense or strong and homogeneous—3) (Figure 3 and Figure 4). Expression levels were compared between the dates of sample collection in specific sections of the intestines. Meaningful differences in the IE of ERα were not observed in the descending colon in the control group and in the ascending colon and descending colon in the experimental group. Meaningful differences in the IE of ERβ were not noted in the caecum and ascending colon in group C, and in the duodenal cap, the third section of the duodenum and the caecum in group E. The intestinal sections where no significant differences were found are not presented graphically. 

On each date of analysis, ERα was more highly expressed in the control group than in the experimental group, especially at absorbance level 0 (Figure 3A–D). Significant differences in ERα expression were found in the control group at different absorption levels, but absorption was significantly more pronounced on dates I, II, and VI. Significant differences in ERα expression were also observed at other absorption levels, but the noted values were much lower than at absorption level 0, and they were only found in the small intestine (Figure 3A–D). In the control group, the average ERα expression was highest at absorbance level 0, and it increased when the digesta entered the caudal segment of the small intestine. 

An analysis of the IE of ERα revealed that it was suppressed in most intestinal segments on all dates in group E (0 points on a 4-point scale), but significant differences were detected only on dates I, II, and VI (Figure 3a). ERα was more highly expressed in the ascending and descending colon at absorption level 3 in the experimental group than in the control group. However, in group E, ERα expression was suppressed at all absorption levels (Figure 3a–d). Differences in the ERα expression were noted in the control group, but only in selected segments of the small intestine, particularly in both parts of the duodenum examined in the study (Figure 3a,b). Similarly to group C, ERα expression was induced in the experimental group at absorbance level 0, whereas at absorbance level 3, the levels of ERα expression in the analysed intestinal segments were higher in the experimental group than in the control group.

In group C, the IE of ERβ was suppressed in both segments of the duodenum, jejunum, and descending colon (Figure 4A–D). The average values of ERß expression in the control group and in the experimental group followed a certain trend. In group E, ERß expression was observed at absorbance level 3, and ERβ was more strongly expressed in all analysed tissues, but its expression was more suppressed at absorbance level 0. However, these differences were not significant. An immunohistochemical analysis of ERß expression in the examined intestinal segments, compared with ERα expression, revealed completely different results. In group E, ERβ was more strongly expressed, especially at absorption level 3 and, interestingly, in the jejunum and colon (Figure 4a–c). However, significant differences between the groups were found only on dates I, II, and III, especially in the examined segments of the duodenum, which can be explained by the fact that ERβ saturation was lower in the duodenum than in the other intestinal segments.

### 2.3. The Prognostic Value of the ERs Expression Profile 

A total of 432 samples were analysed to determine the ER expression indicator (P-ERs). In many of the analysed samples, there were no significant differences in ER expression. The mean values of P-ERs were 42 ± 27 for ERα and 38 ± 26 for ERβ. P-ERs values were not normally distributed (Table 1).

#### 2.3.1. P-ER Values for ERα

In group C, the P-ERα value was 42, reaching 15 in the lower quartile and 69 in the upper quartile. An analysis of the median and the upper and lower quartiles revealed that the expression values could be divided into four subgroups: A—very low P-ERα (P-ERα <15), B—low P-ERα (15 ≤ P -ERα < 42), C—high P-ERα (42 ≤ P-ERα < 69) and D—very high P-ERα (P-Erα ≥69) (Table 1). In group C, very low (A), low (B), high (C), and very high (D) P-ERα values were found in 11 (46%), seven (29%), five (21%), and one (4%) cases, respectively. The statistical analysis was conducted for different means, medians, upper and lower quartiles of the separation points, but no meaningful differences were observed.

The results of the analyses involving the uptake of only Erα or ERβ are difficult to interpret. The values of P-ERs (Table 1) provide new information on the presence of a low ZEN dose in the diet. These were very similar in both groups, but at absorption level 3, an increase in P-ERs was observed in group E, resulting in a shift from quartile A to quartile B from the jejunum directly to the descending colon. The results described above and previous research findings suggest that ZEN may compensate for E_2_ deficiency by triggering ERα [27].

#### 2.3.2. P-ER Values for ERβ

In group C, the P-ERβ worth was 35, reaching 9 in the lower quartile and 61 in the upper quartile. Based on the average value of the median, and the upper and lower quartiles, expression values were divided into four subgroups: A—very low P-ERβ (P—ERβ ≤ 9), B—low P-ERβ (9 ≤ P-ERβ < 35), C—high P-ERβ (35 ≤ P-ERβ < 61), and D—very high P-ERβ (P-Erβ ≥ 61) (Table 1). In group C, very low (A), low (B), high (C), and very high (D) levels of P-ERβ were found in six (25%), 11 (46%), 6 (25%), and one (4%) cases, respectively. The statistical analysis was carried out for different means, medians, upper and lower quartiles, but no meaningful differences were found.

In group E, the P-ERβ value was 38, reaching 12 in the lower quartile and 64 in the upper quartile. Based on the values of the median, the upper and lower quartiles and expression values were divided into four subgroups: A—very low P-ERβ (P-Rβ < 12), B—low P-ERβ (12 ≤ P-ERβ < 38), C—high P-ERβ (38 ≤ P-ERβ < 64) and D—very high P-ERβ (P-Erβ ≥ 64) (Table 1). In the experimental group, very low (A), low (B), high (C), and very high (D) P-ERβ values were known in eight (33%), 10 (42%), three (12%), and three (12%) cases, respectively. The statistical analysis was carried out for different means, medians, upper and lower quartiles, but no meaningful differences were found. 

The values of P-ERβ (Table 1) shifted to the right from quartile C to quartile D at absorption level 3 in the caecum and the ascending colon. An analysis of the expression of both receptors demonstrated that the P-ERα levels shifted significantly to the lower quartiles (to the left) in animals exposed to low ZEN doses.

## 3. Discussion

This study confirmed our recent observations that low ZEN doses improve somatic [36] and reproductive health (our previous mechanistic studies) [2,19,37]. On the first day of exposure, ZEN exerted a stimulatory effect on the body, with the exception of the reproductive system [18,38]. This effect was minimised after the second or third day of exposure, probably due to: (i) the negative effects of extragonadal compensation for oestrogen synthesis [39,40] by androgen conversion or the acquisition of exogenous oestrogens or oestrogen-like substances [2,9,41]; (ii) adaptive mechanisms [37]; (iii) higher energy and protein utilisation, indicating more efficient feed conversion (productivity in group E) [41,42,43]; or (iv) detoxification processes (biotransformation) [3]. The last argument is difficult to confirm since an analysis of the carry-over factor in the GI tract of the same animals did not reveal the inherence of α- ZEL or β- ZEL (ZEN metabolites) in the intestinal walls or that the registered levels were below the detection limit [20,25]. According to López-Calderero et al. [44], a higher ERα/ERβ ratio indicates that proliferative processes are stimulated or silenced, and it is unrelated to apoptosis [38]. Similar observations were made by Cleveland et al. [45] and Williams et al. [46]. These results suggest that low levels of ZEN in the diet stimulate proliferative processes in the gastrointestinal tract of prepubertal gilts, especially in the colon. In sexually mature animals, this is a good predictor of weight gain or the time needed to reach slaughter weight [41], and it suggests that the gastrointestinal tract regulates somatic health [9,38]. Thus, the digestive system acts as a “second brain” [47] as it performs numerous functions including a modulatory role between the intestinal contents and tissues vis. the central nervous system [48]. These findings also suggest that ZEN and endogenous oestrogens control growth, differentiation and other important functions in tissues including in the gastrointestinal tract [2] of prepubertal gilts with supraphysiological oestrogen levels [18]. The above also suggests that oestrogen signalling (e.g., ZEN and its metabolites), regardless of its origin, is the major regulator of genomic mechanisms. Oestrogen receptors play a special role: (i) they are activated by ligand-dependent and ligand-independent pathways; (ii) they act as transcription factors that activate and trigger the expression of all sensitive genes; and (iii) the feedback loop regulated by oestrogens contributes to the maintenance or modification of all genomic processes.

### 3.1. Oestrogen Receptors

The biological effects of oestrogens are determined by the type of ERs including the classical nuclear ERα and ERβ as well as the G-protein-coupled ERs (GPER; its expression has not been analysed). Therefore, the levels of different ERs determine the effects of endogenous and exogenous oestrogens on cells (tissues). 

#### 3.1.1. Oestrogen Receptor Alpha 

The expression of ERα in the control group could be attributed to the physiological deficiency of E_2_ in the gilts before puberty [4,24,49], which could point to supraphysiological hormone levels rather than hypoestrogenism [18,50]. Zearalenone mycotoxicosis contributes to an increase in steroid levels (endogenous steroids such as E_2_, progesterone, and testosterone as well as exogenous steroids such as ZEN), which may restore or enhance ER signalling in cells [18,51], but only in relation to hormone-dependent ERs [27]. As a result, ERα expression is not stimulated but deregulated [51]. Most importantly, circulating steroid hormones are bioavailable (not bound to carrier proteins) and their cellular effects are observed at very low concentrations of approximately 0.1–9 pg/mL E_2_ [49]. The concentrations of active hormones are determined by the age and health status of animals [2,8,18,24,52].

Various conclusions can be drawn from the observations of the role of ERα in mammals and the results of the experimentally induced ZEN mycotoxicosis. According to Suba [38], both high and low levels of E_2_ stimulate the expression and transcriptional activity of ERs to restore or enhance ER signalling in cells, which was not observed in the current study. However, the IE of ERα was suppressed to a greater extent. Low ZEN doses in the diet decrease the IE of ERα, which directly affects the somatic (higher weight gain) [41] and reproductive health (delayed sexual maturity [53]) of animals. It should also be noted that low serum E_2_ levels may induce compensatory effects to increase the expression and transcriptional activity of ERs, while increased synthesis of endogenous E_2_ may compensate for low ER signalling [54]. However, it remains uncertain as to whether low ZEN doses are sufficient to meet the requirements of sexually immature gilts. The present findings suggest that this may be the case, with positive implications for pig farmers. 

#### 3.1.2. Oestrogen Receptor Beta

According to the literature, intense ERβ expression or a high level of absorption (3 points on a 4-point grading scale) contributes significantly to gut health, especially colon health, and intensifies metabolic processes [55,56]. In turn, ERβ silencing increases the risk of duodenal inflammation and enhances oncogenesis not only in the gastrointestinal tract, but also in the reproductive system [22,40,45,46,57]. Deletion processes suggest that ERβ has anti-inflammatory and anti-carcinogenic properties, and exerts chemopreventive effects in the colon [58], which was confirmed in a study of low-dose ZEN mycotoxicosis [59]. 

Apart from the previously published research on the effects of E_2_ deficiency in prepubertal animals, another issue should be addressed. Williams et al. [46] and Gajęcka et al. [59] reported that selected phytoestrogens (silymarin and silibinin) and mycoestrogens (ZEN) have a selective affinity for ERβ [60,61]. This is the result of the increased expression of the ERβ gene, suggesting that natural exogenous dietary oestrogens may have anti-inflammatory properties [35]. These oestrogens also exert chemopreventive effects [22], and they can reverse minor carcinogenic changes in the colon [62]. Calabrese et al. [63] found that a mixture of phytoestrogens and lignans reduced the size and number of duodenal polyps and exerted therapeutic effects in this segment of the gastrointestinal tract [64].

As stated in the research objective, this study was conducted to determine if low ZEN doses naturally occurring in feeds could produce similar effects, and the present results suggest that it is possible. This conclusion is also consistent with the results of previous studies conducted as part of the same research project [2,19,29,31,32,33,34,35,36,52].

#### 3.1.3. ER Expression Indicator

In animals exposed to ZEN, the P-ER levels differed between quartiles. In group E, the P-ERα values shifted from quartile A to quartile B, while the P-ERβ values shifted from quartiles B and C to quartiles A and D. The expression levels of ERα confirm that low ZEN doses can exert oestrogenic effects on the studied ERs. 

The endogenous ligand that triggers ERβ [27] and the cells that are activated by specific receptors could not be identified based on the existing knowledge. For this reason, the influence of ZEN on ERβ is difficult to interpret. It seems that E_2_ does not bind to ERα and ERβ with equal affinity, but it binds to oestrogen response elements. However, ERβ is a much weaker transcriptional activator than ERα. In turn, the oestrogen response element activator protein-1 is responsible for the proliferation processes induced by E_2_. Nevertheless, E_2_ has no effect on ERβ, which may indicate that ERβ can modulate ERα activity in cells where both receptors are co-expressed. However, in many cells, ERβ is expressed in the absence of ERα, and in these cells, ERβ remains active independently of ERα [56]. This is the case in epithelial cells of the colon [65], where ERβ-driven enhanced metabolic processes occur [55].

Preclinical models have shown that ERα activity can be modulated by ERβ, which inhibits oestrogen-dependent proliferation and promotes apoptosis [66]. There is evidence that uncontrolled proliferation, progression, and/or failure to respond to treatment may disrupt oestrogen signalling. ERα may be associated with proliferative disorders, and it can be used to determine the efficacy of hormone therapy. In contrast, ERβ is present in healthy colonic mucosa and its expression is significantly delayed in colonic proliferative disorders [44,56].

#### 3.1.4. Summary

The observed silencing of ERs indicates that: (i) low monotonic doses of ZEN elicited the strongest responses on analytical dates III, IV, and VI, whereas on the last date, the prepubertal gilts developed tolerance to the analysed undesirable substance; (ii) ERα expression was increased in the duodenum and ERβ expression was increased in the descending colon; (iii) the opposite was observed in the caecum and the ascending colon; and (iv) the gastrointestinal tract of sexually immature gilts was adapted to the presence of ZEN in the feed after the first two exposure dates. Due to the very low concentrations of E_2_, ZEN was bound to ERs and triggered qualitative changes in ERs during the successive weeks of the experiment (activation?). Qualitative changes were manifested by a shift in the ER expression levels from absorption level 0 to 3, especially ERβ expression in the descending colon. The observed shift in ERβ expression suggests that zearalenone and its metabolites are involved in the control of proliferation and apoptosis in enterocytes.

## 4. Materials and Methods

### 4.1. Experimental Animals

The experiment was carried out at the Department of Veterinary Prevention and Feed Hygiene of the Faculty of Veterinary Medicine of the University of Warmia and Mazury in Olsztyn, Poland, on 36 clinically healthy gilts with an initial body weight (BW) of 25 ± 2 kg. Pre-puberty gilts were kept in groups and had ad lib access to water. 

### 4.2. Experimental Feed

The feed administered to animals (Table 2) was analysed for the presence of ZEN and DON. Mycotoxin content was determined by standard separation techniques using immunoaffinity columns (Zearala-TestTM Zearalenone Testing System, G1012, VICAM, Watertown, MA, USA; DON-TestTM DON Testing System, VICAM, Watertown, MA, USA) and high-performance liquid chromatography (HPLC) (Hewlett Packard, type 1050 and 1100) [67] with fluorescence and/or ultraviolet detection techniques. The detection limit was 3.0 ng/g for ZEN [19] and 1.0 ng/g for DON [36].

### 4.3. Experimental Design

The animals were allocated to an experimental group (E = ZEN; n = 18) and a control group (C, n = 18) [68,69]. The animals in group E were orally administered ZEN at a dose of 40 μg/kg BW (Table 3). The pigs in group C were given a placebo. At the time when this test was designed, the above value complied with the recommendations of the European Food Safety Authority (CR 2006/576/EC—2006 [70]) and No-Observed-Adverse-Effect Level (NOAEL) dose. The mycotoxin was administered every morning before feeding, in gel capsules that dissolved in the stomach. In group C, pigs received identical gel capsules, but without the mycotoxin. 

Zearalenone was biosynthesised at the Faculty of Chemistry at the University of Life Sciences in Poznań. The trial lasted 42 days. Zearalenone doses were adapted to the BW of gilts. Zearalenone was served in capsules to avoid potential problems resulting from unequal feed intake. Zearalenone samples were dissolved in 500 μL 96% C_2_H_5_OH (96% ethyl SWW 2442-90, Polskie Odczynniki Chemiczne SA, Poland) to obtain the required dose (converted to BW). The solutions were kept at 20 °C for twelve hours. The gilts were weighed at weekly intervals to adjust the ZEN dose of each animal. Three gilts from each group (six animals in total) were euthanised on days 7 (date I), 14 (date II), 21 (date III), 28 (date IV), 35 (date V), and 42 (date VI) by intravenous administration of sodium pentobarbital (Fatro, Ozzano Emilia BO, Italy). Directly after cardiac arrest, part of the intestinal tissue were taken and prepared for analysis.

### 4.4. Reagents 

ZEN was obtained from the Faculty of Chemistry, University of Life Sciences in Poznań based on an earlier developed methodology [71,72] presented in other studies [73].

### 4.5. Chemicals and Equipment

The chromatographic analysis of ZEN was conducted at the Faculty of Chemistry, University of Biosciences in Poznań based on an earlier developed methodology [73].

### 4.6. Tissue Samples

On each experimental day, intestinal tissue samples (approx. 1 × 1.5 cm) were collected from the succeeding segments of the GI tract of gilts: the duodenum—the first part and the third section; the jejunum and ileum—the middle part; the large intestine—the middle parts of the ascending colon, transverse colon and descending colon; and the caecum—1 cm from the ileocecal valve. The samples were rinsed with phosphate buffer.

### 4.7. Immunohistochemistry 

#### 4.7.1. Localisation of ERα and ERβ 

Tissue samples were fixed in four percent paraformaldehyde and embedded in paraffin. Two samples from each test section were stained to determine the ERα and ERβ expression. In the negative control, the primary antibody was omitted. To unmask the antigens, the sections were placed in citrate buffer (Sigma-Aldrich, Saint Louis, MO, USA) and cooked for 20 min in a microwave oven at 800 W. The sections were coated with ready-to-use DAKO REALTM Peroxidase Blocking Solution (DAKO, Glostrup, Denmark) and reacted for 15 min. Non-specific antigen binding areas were blocked with 2.5% normal goat serum solution. The sections were reacted overnight at a temperature of 6 °C with the following primary antibodies: Mouse Anti-Human Oestrogen Receptor α (Clone: 1D5, DAKO Santa Clara, CA, USA) and Mouse Anti-Oestrogen Receptor β (Clone: 14C8, Abcam, Cambridge, UK), diluted to 1:60 and 1:20, respectively. After the reaction, the specimens were rinsed three times with PBS (Sigma-Aldrich, Saint Louis, MO, USA) at five-minute intervals. Secondary antibodies conjugated with horseradish peroxidase-labelled micropolymer (ImmPRESS™ HRP Universal Antibody, Vector Laboratories, Burlingame, CA, USA) were applied to the specimens. The sections were coloured by incubation with DAB (DAKO, Glostrup, Denmark) for 3 min, and H_2_O_2_ was added to visualise the activity of the bound enzyme (brown colour). The sections were washed with water and contrast stained with Mayer’s haematoxylin solution (Sigma-Aldrich, Saint Louis, MO, USA). The primary antibody was ignored in the negative control. Negative controls (solvent-coated slides only, no primary antibody) and positive controls were converted together with the slides [74]. The pig’s ovary was used as a positive control for ERβ [75]. 

#### 4.7.2. Scanning of the Coloured Slides

The expressions of ERα and ERβ were analysed on the scanned slides (Pannoramic MIDI scanner, 3DHISTECH, Budapest, H) using the NuclearQuant programme (3DHISTECH, H). The slides were converted into digital images (Figure 1 and Figure 2). The profile of nuclear detection and staining intensity were as previously described [59].

### 4.8. Statistical Analysis

The activity of ERα and ERβ in the GI tract of pigs was presented on the basis of ± and SD for each sample. The results were compiled using the Statistica programme (StatSoft Inc., USA). Based on the applied ZEN dose and the duration of its application, the arithmetic means for systems with repeatable measurements were compared using one-way analysis of variance. The homogeneity of variance in the compared groups was checked with the Brown–Forsythe test. Differences between groups were analysed using Tukey’s honestly significant difference test (*p* < 0.05 or *p* < 0.01).

## Figures and Tables

**Figure 1 toxins-15-00122-f001:**
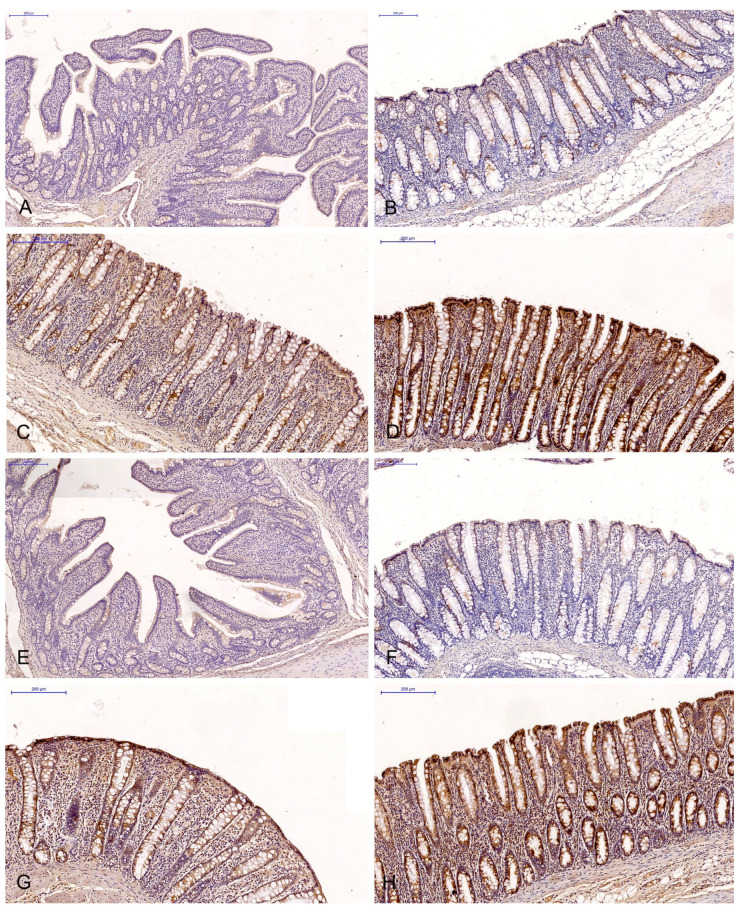
Scanned slides showing the IE of ERα in the descending colon in group C ((**A**)—0; (**B**)—+; (**C**)—++; (**D**)—+++) and group E ((**E**)—0; (**F**)—+; (**G**)—++;. (**H**)—+++). HE.

**Figure 2 toxins-15-00122-f002:**
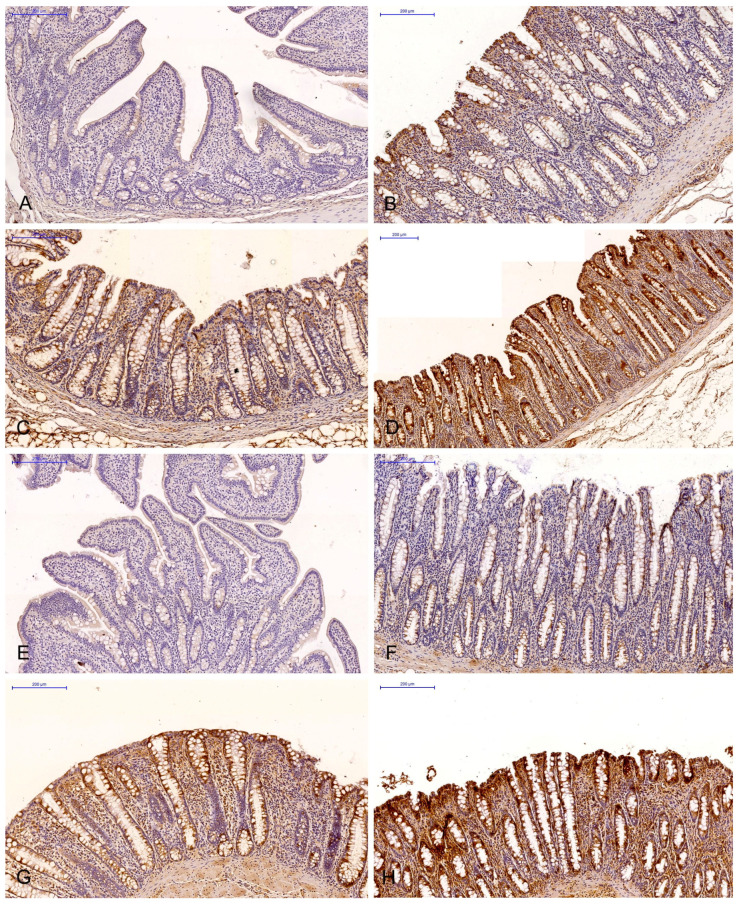
Scanned slides showing the IE of ERβ in the descending colon in group C ((**A**)—0; (**B**)—+; (**C**)—++; (**D**)—+++) and group E ((**E**)—0; (**F**)—+; (**G**)—++; (**H**)—+++). HE.

**Figure 3 toxins-15-00122-f003:**
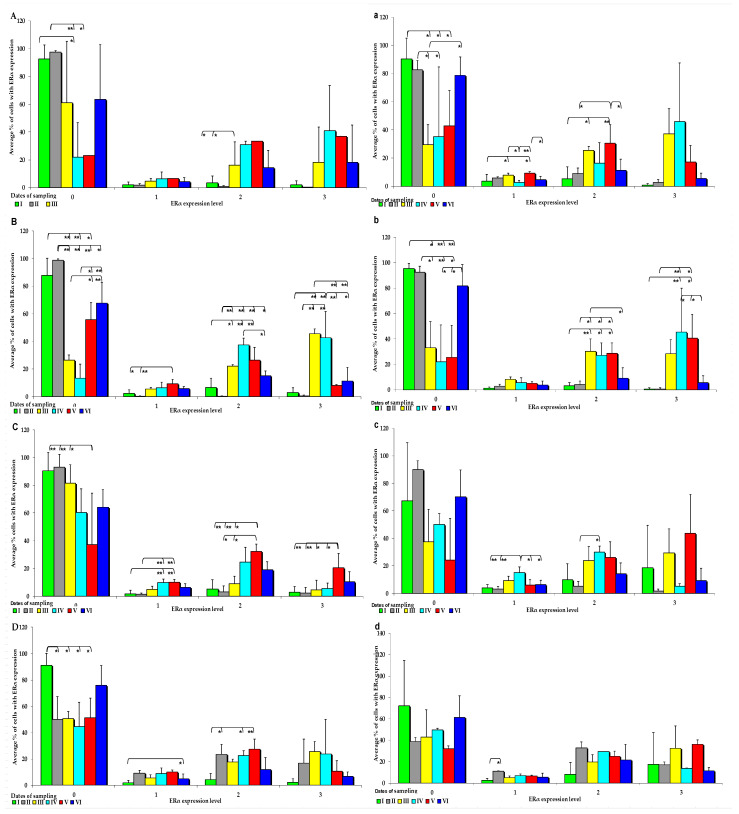
IE of ERα (based on a 4-point grading scale: negative—0; weak and homogeneous—1; mild or moderate and homogeneous—2; intense or strong and homogeneous—3) in the intestines of sexually immature gilts from the control group: (**A**) in the duodenal cap on selected dates of exposure; (**B**)—in the third section of the duodenum on selected dates of exposure; (**C**) in the jejunum on selected dates of exposure; (**D**) in the caecum on selected dates of exposure. In the intestines of sexually immature gilts from the experimental group: (**a**) in the duodenal cap on selected dates of exposure; (**b**) in the third section of the duodenum on selected dates of exposure; (**c**) in the jejunum on selected dates of exposure only in the weak(1) and mild (2) grades; (**d**) in the caecum on selected dates of exposure only in the weak grade (1). Expression was presented as ± (confidence interval) and SE (standard error) for some samples. * *p* ≤ 0.05 and ** *p* ≤ 0.01 compared with the residual groups.

**Figure 4 toxins-15-00122-f004:**
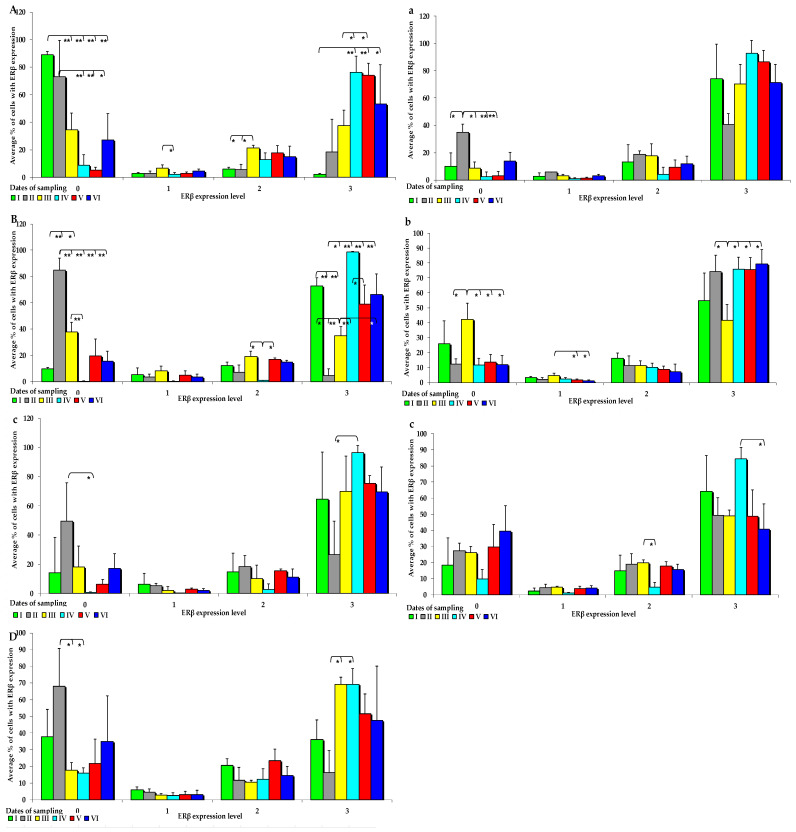
IE of ERβ (based on a 4-point grading scale: negative—0; weak and homogeneous—1; mild or moderate and homogeneous—2; intense or strong and homogeneous—3) in the intestines of sexually immature gilts from the control group: (**A**) in the duodenal cap on selected dates of exposure; (**B**) in the third section of the duodenum on selected dates of exposure; (**C**) in the jejunum on selected dates of exposure only in the negative (0) and intense (3) grades; (**D**) in the descending colon on selected dates of exposure only in the negative (0) and intense (3) grades; in the intestines of sexually immature gilts from the experimental group: (**a**) in the jejunum on selected dates of exposure only in the negative grade (0); (**b**) in the ascending colon on selected dates of exposure; (**c**) in the descending colon on selected dates of exposure only in the mild (2) and intense (3) grades. Expression was presented as ± (confidence interval) and SE (standard error) for some samples. * *p* ≤ 0.05 and ** *p* ≤ 0.01 compared with the residual groups.

**Table 1 toxins-15-00122-t001:** ERα and ERß expression at various absorption levels in the analysed sections of the GI tract in pre-pubertal gilts.

Group	Absorption	Duodenal Cap	Third Part of Duodenum	Jejunum	Caecum	Ascending Colon	Descending Colon
ERα							
Group C	0	C	C	C	C	C	D
	1	A	A	A	A	A	A
	2	B	B	B	B	B	A
	3	B	B	A	A	A	A
Group E	0	C	C	C	C	C	D
	1	A	A	A	A	A	A
	2	B	B	B	B	B	B
	3	B	B	B	B	B	B
ERβ							
Group C	0	C	B	B	B	B	B
	1	A	A	A	A	A	A
	2	B	B	B	B	B	B
	3	C	C	D	C	C	C
Group E	0	B	B	B	B	B	B
	1	A	A	A	A	A	A
	2	B	B	B	A	A	B
	3	C	C	D	D	D	C

Abbreviation: In group E, the value of P-ERα was 35, reaching 8 in the lower quartile and 62 in the upper quartile. The analysed expression values were divided into four subgroups based on the values of the median, and the upper and lower quartiles: A—very low P-ERα (P-Erα ≤ 8), B—low P-ERα (8 ≤ P-Erα < 35), C—high P-ERα (35 ≤ P-Erα < 62), and D—very high P-ERα (P-Erα ≥ 62) (Table 1). In group E, very low (A), low (B), high (C), and very high (D) values of P-ERα were noted in six (25%), 12 (50%), five (21%), and one (4%) cases, accordingly. The statistical analysis was carried out for different mean, median, upper and lower quartile cut-off points, but no meaningful differences were noted.

**Table 2 toxins-15-00122-t002:** Mixture of diets for pre-pubertal gilts (first stage of rearing).

Percentage Content of Feed Ingredients		Nutritional Value of Diets	
Barley (*Hordeum* L.)	27.65	Metabolizable energy MJ/kg	12.575
Wheat (*Triticum monococcum* L.)	17.5	Total protein (%)	16.8
Triticale (*Triticosecale* Wittm. ex A.Camus)	15.0	Digestible protein (%)	13.95
Maize (*Zea mays* L.)	17.5	Lysine (g/kg)	9.975
Soybean meal, 46%	16.0	Methionine + Cysteine (g/kg)	6.25
Rapeseed meal	3.5	Calcium (g/kg)	8.05
Limestone	0.35	Total phosphorus (g/kg)	5.75
Premix ^1^	2.5	Available phosphorus (g/kg)	3.1
		Sodium (g/kg)	1.5

Abbreviation: Composition of the vitamin-mineral premix per kg: vitamin A—500.000 IU; iron—5000 mg; vitamin D3—100.000 IU; zinc—5000 mg; vitamin E (alpha-tocopherol)—2000 mg; manganese—3000 mg; vitamin K—150 mg; copper (CuSO_4_·5H_2_O)—500 mg; vitamin B1—100 mg; cobalt—20 mg; vitamin B2—300 mg; iodine—40 mg; vitamin B6—150 mg; selenium—15 mg; vitamin B12—1500 μg; niacin—1200 mg; pantothenic acid—600 mg; L-threonine—2.3 g; folic acid—50 mg; tryptophan—1.1 g; biotin—7500 μg; phytase + choline—10 g; ToyoCerin probiotic + calcium—250 g; magnesium—5 g.

**Table 3 toxins-15-00122-t003:** Diurnal feed intake in a restricted feeding regime (kg/day) and the average zearalenone concentration per kg feed (μg ZEN/kg feed).

Week of Exposure	Feed Intake	Total ZEN Dose	
	kg/Day	µg ZEN/kg BW	µg ZEN/kg Feed
I	1.1	280	1014
II	1.0	560	972
III	1.3	840	1014
IV	1.6	1120	987
V	1.9	1400	995
VI	1.7	1680	957

## Data Availability

Not applicable.

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
