# Peer review of "Immunohistochemical Expression (IE) of Oestrogen Receptors in the Intestines of Prepubertal Gilts Exposed to Zearalenone"

_toxins, 2023, doi:10.3390/toxins15020122_

Round 1
Reviewer 1 Report
Review of toxins-2143614
This paper describes changes in oestrogen receptor α and β with low dose Zearalenone in different regions of the intestine over time in prepubertal gilts using immunohistochemistry.
Introduction:
Please indicate how the low dose of 40 μg/kg bw relates to what gilts are commonly exposed to under normal husbandry conditions; relate this to concentrations normally used under experimental conditions.
Results:
Figures 1 & 2 Line 81 to 84. The statement about non-specific staining is confusing. Do the authors mean all the staining is non-specific? Please clarify. A negative control image should be included in each figure.
Line 96 – 98. It is unclear what is being compared to provide significant or non-significant results. Is this staining intensity scale in the different regions of the intestine? How does this statement relate to time point sampled? How does this relate to individual gilts sampled?
Line 108. It is unclear what “+/- and SD” indicates
Figures 3 and 4. It is unclear what the y axis % is. Is this percent of the three samples taken at each time point that staining level 0-3? Is it the average staining intensity of the three samples at each time point? How can there be an average if all the sections from one region of the intestine at that time point scored a 0?
All the figures 3 & 4, and 5 & 6 would be better presented to show both groups C and E in the same figure side by side. This should be done for all regions of the intestine rather than making the reader jump between figures to see differences with treatment. This would provide a direct visualization of treatment and control. It would also be helpful to include the non-significant regions of the intestine as well to provide a “view” of changes across the whole intestinal tract rather than the piecemeal presentation. This is compounded with different regions of the intestine shown in each figure. The current presentation makes it impossible to directly compare exposed and control groups.
Line 130. Higher expression in group E compared to group C is mentioned for ascending colon and descending colon yet neither of these regions are shown in figure 3 or 4 nor are data shown.
Line 156. States that expression is suppressed but it is unclear what it is compared to.
Line 160. How can expression be more or less expressed if there is no significant difference?
This section gives a subjective assessment between groups E and C at different time points. There is no statistical comparison given so one does not know if these observations are meaningful. Any comparison between treatments needs to be analyzed with a two-way ANOVA. There are two dependent variables in this study – time and treatment with the independent variable being staining intensity. These variables cannot be viewed independently with only a one-way ANOVA
Section 2.4.1 and Table 1. These data cannot be statistically compared by conflating three variables – in this case: time, treatment, and staining intensity. This analysis requires a 3 way ANOVA as there are three variables.
Methods
The source and type of antibodies used for the immunohistochemistry are not given.
Many of the methods are given just as a citation. This is fine, but in some circumstances a brief sentence describing critical points should be given along with the reference to allow the reader to understand the appropriateness of the methods.
Discussion
It is not clear (when the majority of results were not statistically significant) how the results support the analysis in the discussion.
Line 221. The statement “The results of the present study confirm our recent observations that low ZEN improve somatic and reproductive health (our previous mechanistic studies)” is not substantiated. How do the new results confirm the existing data? More detail is needed to support this statement.
The remainder of the discussion relates to the effect of low dose ZEN on reproduction but other than the statement above there is no discussion of how the present results relate to this existing body of knowledge.
Author Response
I Reviewer
Review of toxins-2143614
This paper describes changes in oestrogen receptor α and β with low dose Zearalenone in different regions of the intestine over time in prepubertal gilts using immunohistochemistry.
Introduction:
Please indicate how the low dose of 40 μg/kg bw relates to what gilts are commonly exposed to under normal husbandry conditions; relate this to concentrations normally used under experimental conditions.
Dear reviewer – this is due to publicly available materials published by EFSA. In the chapter "Material and Methods" subsection "4.3. Experimental design" Lines 363-365 – we justify why weused this dose. This justification is based on a published suggestion on ZEN doses (Commission Recommendation 2006/576/EC – [70]). Jest is the no-observed-adverse-effect level (NOAEL), i.e. a dose that does not yet cause clinical symptoms [Lines 62-65]. Due to the fact that ZEN is an undesirable substance and is normally found in plant material, so contact with it is widespread. We have not presented preliminary data on this mycotoxin, because the topic itself is very extensive. Therefore, we would be very grateful if we were exempted from the obligation to enter this type of information in this work.
Results:
Figures 1 & 2 Line 81 to 84. The statement about non-specific staining is confusing. Do the authors mean all the staining is non-specific? Please clarify. A negative control image should be included in each figure.
Dear reviewer – the different levels expression of ERα and ERs were observed in all evaluated segments of the intestine (Figs. 1 and 2). The negative controls were performed each time during staining to avoid nonspecific staining. The term “non-specific staining” was used here erroneously and whole part of this section was rewritten.
Corrected section: The brown background staining of the scanned slides (Figures 1 and 2) was not specific to all intestinal segments and may have occurred in staining assays examining ERα and ER expression in DAB-stained gastrointestinal tissues (non-specific light brown staining was observed in most samples).
Line 96 – 98. It is unclear what is being compared to provide significant or non-significant results. Is this staining intensity scale in the different regions of the intestine? How does this statement relate to time point sampled? How does this relate to individual gilts sampled?
Dear reviewer – after reading the comment and re-examining our text, I must conclude that the indicated passage is badly or misleadingly written. Therefore, we propose to introduce an additional sentence into the existing description. It reads: The expression values were statistically compared between the dates of collection in specific sections of the intestines.
Line 108. It is unclear what “+/- and SD” indicates
Dear reviewer – information about the meaning of the presented symbols (±) Confidence Interval and (SD) Standard Deviation has been supplemented in the text of the work.
Figures 3 and 4. It is unclear what the y axis % is. Is this percent of the three samples taken at each time point that staining level 0-3? Is it the average staining intensity of the three samples at each time point? How can there be an average if all the sections from one region of the intestine at that time point scored a 0?
Dear Reviewer – we apologize for not makingthe appropriate provision. The y-axis is "Average % of cells with ERs expression". All Figures have been supplemented with this notation.
All the figures 3 & 4, and 5 & 6 would be better presented to show both groups C and E in the same figure side by side. This should be done for all regions of the intestine rather than making the reader jump between figures to see differences with treatment. This would provide a direct visualization of treatment and control. It would also be helpful to include the non-significant regions of the intestine as well to provide a “view” of changes across the whole intestinal tract rather than the piecemeal presentation. This is compounded with different regions of the intestine shown in each figure. The current presentation makes it impossible to directly compare exposed and control groups.
Dear reviewer – we have fulfilled all the suggestions. Now there are only Figures 3 and 4.
Line 130. Higher expression in group E compared to group C is mentioned for ascending colon and descending colon yet neither of these regions are shown in figure 3 or 4 nor are data shown.
Dear Reviewer – we did not present these items graphically because there were no statistical differences, which is indicated in the relevant section of Lines 96-101. The presented fragment was supplemented with the appropriate paragraph – "Intestinal sections in which no statistical differences were found are not presented graphically. "
Line 156. States that expression is suppressed but it is unclear what it is compared to.
Dear reviewer – the presented suggestion is true, but it is group C so it can be taken as a kind of norm. We are not able to explain all the situations that have occurred.
Line 160. How can expression be more or less expressed if there is no significant difference?
Dear reviewer - we would like to point out that a certain degree of expression (some) always exists. We have learned to appreciate the importance of differences only if the differences are statistical, but when it concerns (statistically insignificant) individuals, we ignore them, even though there is expression.
This section gives a subjective assessment between groups E and C at different time points. There is no statistical comparison given so one does not know if these observations are meaningful. Any comparison between treatments needs to be analyzed with a two-way ANOVA. There are two dependent variables in this study – time and treatment with the independent variable being staining intensity. These variables cannot be viewed independently with only a one-way ANOVA
Dear reviewer – we would like to point out to the honourable reviewer that we made this type of choice deliberately because in our opinion dates and exposure are independent variables. However, theintensity of staining, which was transformed into a digital image and only counted, is a dependent variable and it is evaluated. In other words, with constant exposure of ZEN (lines 362-363) and specific sampling dates (Line 377-379) different ERs expression values take place. In this situation, a one-way ANOVA is sufficient despite the presence of three factors. Each statistical study was performed separately.
Section 2.4.1 and Table 1. These data cannot be statistically compared by conflating three variables – in this case: time, treatment, and staining intensity. This analysis requires a 3 way ANOVA as there are three variables.
Dear Reviewer – we would like to point out that of the three factors mentioned by the reviewer, two are constant (timing of sampling or dose size – ZEN / kg BW), the only variable factor is the value of expression. Therefore, we propose to leave the provision unchanged. On the other hand, these factors can be compared separately each with each. It is not necessary to compare these factors together.
Methods
The source and type of antibodies used for the immunohistochemistry are not given.
Dear Reviewer – the one expected by the Reviewer has been supplemented in the subsection "4.7.1. Localisation of ERα and Erβ" (Lines 386-405).
Many of the methods are given just as a citation. This is fine, but in some circumstances a brief sentence describing critical points should be given along with the reference to allow the reader to understand the appropriateness of the methods.
Dear reviewer – If it is possible, we have only supplemented the subsection "4.7.1. Localisation of ERα and Erβ". We suggest not inserting the other two due to the fact that these are technical methods and from the point of view of the so-called anti-plagiarism control are very cumbersome in editorial correction.
Discussion
It is not clear (when the majority of results were not statistically significant) how the results support the analysis in the discussion.
Dear reviewer – this is interesting, because approaching the subject from a physiological point of view, clusters of ERs in young maturing organisms (especially females) take place mainly in the colon. We examined the remaining sections of the intestines not out of conviction but to confirm the already known facts. And ZEN has no influence on this. What remains is the colon, which gave statistical differences in the expression of ERs and that satisfied us. Such information is included in the zearalenone background, which we have not presented in this paper as suggested by the publisher. Similar analyses on the effect of deoxynivalenol and mixed mycotoxicosis on ERs expression are presented in two other recognized scientific reports https://doi.org/10.1016/j.toxicon.2019.11.006 and doi: 10.3389/fvets.2021.644549].
Line 221. The statement “The results of the present study confirm our recent observations that low ZEN improve somatic and reproductive health (our previous mechanistic studies)” is not substantiated. How do the new results confirm the existing data? More detail is needed to support this statement.
Dear reviewer – the cited papers in the introduction to the Discussion chapter document this sentence and allowed us to extrapolate the currently presented results to the results presented earlier where the effect of exposure with much lower doses was examined (5, 10 and 15 μ g ZEN/ kg BW prepubertal gilts). Once again, I would like to point out that the results obtained now and earlier allow only a comparison in the form of extrapolation and not a direct comparison. Secondly, ZEN is a mycoestrogen (Lines 291-294) – therefore, most studies deal with problems related to tissues where ERs are located.
The remainder of the discussion relates to the effect of low dose ZEN on reproduction but other than the statement above there is no discussion of how the present results relate to this existing body of knowledge.
Dear reviewer – the problem is that our knowledge of the health significance of zearalenone mycotoxicosis provoked by high doses of exposure (100 μg ZEN/kg BW) is significant. On the other hand, the effect of exposure with doses of NOAEL or MABEL is negligible. The symptoms accompanying subclinical conditions are usually unknown to us and are very difficult to define and interpret. Due to such a small amount of knowledge, it is difficult to achieve the results obtained. An extrapolation has been done on the subject and it looks like this at the moment. It should also be remembered that clinical and anatomopathological outcomes of high exposures cannot be directly transferred to low exposure results according to the law of hormesis.

Reviewer 2 Report
The ms toxins-2143614 with the title of Immunohistochemical expression of oestrogen receptors in the intestines of prepubertal gilts exposed to Zearalenone investigates an interesting topic, but the authors have to improve it significantly to make it suitable for such high quality journal.
In abstract, please add the most important findings in terms of some values.
L44-47 Many oestrogen-sensitive tumours are termed … please add relevant citations for these two sentences
L48-49 The chemopreven- … please add relevant citation for these two sentence
L53-54 add for this sentence at least two citations>> The elements of the oestrogen response have been investigated in studies
L56-59 cite this text please
L65-68 Therefore this experiment aimed to study whether… please add the hypothesis of this study at the end of the introduction. Then, put the aim and hypothesis in one independent paragraph at the end of the introduction.
L72-73 The concentrations of masked mycotoxins were not investigated! If was not investigated, then remove it.
Sections 2.1 and 2.2. if the results were not presented in this ms, why the authors want to add this text here.
Figure 1. instead of writing (Figure 1A – etc….. please make it simple as follows: (A –
What I mean, the authors should remove the word Figure 1 because it is repeated many times in the title of Figure 1
Same comment is for title of Figure 2, please revise
Figures 3-6 authors should remove the data that are presented under each graph. The authors should choose to present their data either in Figures or Tables, but avoid presenting both for same data.
The authors should use SE standard error instead of SD standard deviations for the bars on the top of the columns, same for Tables, use SE instead of SD because the error bars are very big.
Remove the values presented above columns, the readers already can know what are these values.
Table 2, please add the Latin names of Barley, Wheat etc. Also, why Barley is in bold?
What was the name of the Experimental design? Please mention it in the material and methods section.
L416 p < 0.05 or p < 0.01 These are wrong writing. They should be p ≤ 0.05 or p ≤ 0.01, check this in whole ms.
Regards,
Reviewer
Author Response
II Reviewer
The ms toxins-2143614 with the title of Immunohistochemical expression of oestrogen receptors in the intestines of prepubertal gilts exposed to Zearalenone investigates an interesting topic, but the authors have to improve it significantly to make it suitable for such high quality journal.
In abstract, please add the most important findings in terms of some values.
Dear reviewer – the appropriate values are inserted in the abstract
L44-47 Many oestrogen-sensitive tumours are termed … please add relevant citations for these two sentences
Dear Reviewer – appropriate literature items have been introduced [14,21] and [2].
L48-49 The chemopreven- … please add relevant citation for these two sentence
Dear reviewer – appropriate literature items have been introduced [1,8].
L53-54 add for this sentence at least two citations>> The elements of the oestrogen response have been investigated in studies
Dear reviewer – appropriate literature items have been introduced [1,2,13,18].
L56-59 cite this text please
Dear reviewer – appropriate literature items have been introduced [8].
L65-68 Therefore this experiment aimed to study whether… please add the hypothesis of this study at the end of the introduction. Then, put the aim and hypothesis in one independent paragraph at the end of the introduction.
Dear Reviewer, everything that the reviewer pays attention to is presented in the last paragraph. So a hypothesis takes place and in the last sentence the goals of the work are presented. Therefore, we propose to leave it that way.
L72-73 The concentrations of masked mycotoxins were not investigated! If was not investigated, then remove it.
Dear reviewer – the indicated fragment has been removed.
Sections 2.1 and 2.2. if the results were not presented in this ms, why the authors want to add this text here.
Dear reviewer – for two reasons: (i) in studies evaluating the effects of ZEN exposure, feed must be free of "natural" mycotoxins; (ii) In subsection 2.2, we document the fact that the presented study is part of larger studies that have already been published almost in their entirety.
Figure 1. instead of writing (Figure 1A – etc….. please make it simple as follows: (A –
Dear Reviewer, the change was made as suggested by the Reviewer.
What I mean, the authors should remove the word Figure 1 because it is repeated many times in the title of Figure 1
Dear Reviewer, the change was made as suggested by the Reviewer.
Same comment is for title of Figure 2, please revise
Dear Reviewer, the change was made as suggested by the Reviewer.
Figures 3-6 authors should remove the data that are presented under each graph. The authors should choose to present their data either in Figures or Tables, but avoid presenting both for same data.
Dear Reviewer – the change was made in accordance with the Reviewer's suggestion – we resigned from tabular presentation of data.
The authors should use SE standard error instead of SD standard deviations for the bars on the top of the columns, same for Tables, use SE instead of SD because the error bars are very big.
Dear reviewer – because we have resigned from tabular presentation of data, we will stick to Standard Deviation due to the fact that the indicated bars are not so large.
Remove the values presented above columns, the readers already can know what are these values.
Dear reviewer – we not only eliminated the values presented above the columns, but also changed the chart type.
Table 2, please add the Latin names of Barley, Wheat etc. Also, why Barley is in bold?
Dear reviewer – Latin names have been supplemented and bold in barley have been eliminated.
What was the name of the Experimental design? Please mention it in the material and methods section.
Dear Reviewer, the name of the project was inserted at the end of the Material and Methods chapter, but the Editorial Board, sending the work to the Reviewers, cuts out this fragment so that the performers are unknown (incognito) and the Reviewers do not know who is who.
L416 p < 0.05 or p < 0.01 These are wrong writing. They should be p ≤ 0.05 or p ≤ 0.01, check this in whole ms.
Dear reviewer – corrections have been made throughout the work.
Regards,
Regards,
Reviewer
This time the authors

Reviewer 3 Report
Zearalenone, as a mycoestrogen, plays an important role in the growth and development of reproductive and non-reproductive tissues. Subclinical symptoms of ZEN mycotoxicosis can cause changes in hormonal signalling when enterocytes in different intestinal segments are exposed to this mycotoxin. Thus, it is of significance to evaluate the the role of zearalenone in the digestive system of gilts.
Howevere, there are some details need to revise:
-Ln 3: There was a extra space in the title.
-Ln 7: What does Group C and Group E means? It should be written in full at Abstract.
-The result should be more detailed in the Abstract.
Figure 1 and 2 need more description and details. For example, ABCDEFGH refer to sections of which parts respectively?
Table 3 need more description. For example, what dose 280μ g/kg bw mean? It should be refer to in Ln 363.
-Please provide the formula for calculating the percentage in Figure 3, 4 and 5
-Ln 345: Why should DON content be determined in feed? Why are the contents of other toxins not measured?
Author Response
III Reviewer
Zearalenone, as a mycoestrogen, plays an important role in the growth and development of reproductive and non-reproductive tissues. Subclinical symptoms of ZEN mycotoxicosis can cause changes in hormonal signalling when enterocytes in different intestinal segments are exposed to this mycotoxin. Thus, it is of significance to evaluate the the role of zearalenone in the digestive system of gilts.
Howevere, there are some details need to revise:
-Ln 3: There was a extra space in the title.
Dear reviewer – this is the result of insubordination of printing. The place has been liquidated.
-Ln 7: What does Group C and Group E means? It should be written in full at Abstract.
Dear reviewer – the shortcomings have been completed.
-The result should be more detailed in the Abstract.
Dear reviewer – selected results have been presented in the summary.
Figure 1 and 2 need more description and details. For example, ABCDEFGH refer to sections of which parts respectively?
Dear reviewer – we have made a shortcut of the entry under Figure 1 and 2. It seems to us that there is no need to add anything more.
Table 3 need more description. For example, what dose 280μ g/kg bw mean? It should be refer to in Ln 363.
Dear reviewer – appropriate corrections havebeen made in the text as well as in Table 3.
-Please provide the formula for calculating the percentage in Figure 3, 4 and 5
Dear reviewer – I give the method of calculating the percentage values – practically it looked like this: three samples were taken from each gilt for testing, three gilts were euthanized at each time; in total, we had nine stains each time in a specific group at a specific date of the study and hence we calculated the average value of pluses or lack thereof, i.e. "-".
-Ln 345: Why should DON content be determined in feed? Why are the contents of other toxins not measured?
Dear Reviewer – this is due to the fact that DON is a fusion mycotoxin and is very often present together with ZEN, which may distort the test results; in turn, preventive studies show that the presence of other mycotoxins is unlikely in our geographical region and if they are, they are in the values of the so-called background.

Round 2
Reviewer 1 Report
The manuscript is much improved. Thank you for making the suggested changes and clarifying areas of the manuscript
Author Response
I am glad that the changes that were made met Your expectations.
Thank You.
Reviewer 2 Report
The ms has been improved, but Figures 3 and 4 should be improved.
Author Response
The suggested table changes have been implemented.
Thank You